# Dacryocystitis: Is Dacryocystorhinostomy Always the Solution?

**DOI:** 10.3390/jcm13175129

**Published:** 2024-08-29

**Authors:** Alexis Mathieu, Stéphanie Baillif, Marie-Noelle Delyfer, Éric Longueville, Valentine Coste-Verdier, Jacques Lagier, Abdulrhman Alrabiah, Arnaud Martel

**Affiliations:** 1Service d’Ophtalmologie, Centre Hospitalier Universitaire de Bordeaux, F-33000 Bordeaux, France; marie-noelle.delyfer@chu-bordeaux.fr (M.-N.D.); elongueville@wanadoo.fr (É.L.); valentine.coste@chu-bordeaux.fr (V.C.-V.); 2Service d’Ophtalmologie, Centre Hospitalier Universitaire de Nice, F-06000 Nice, France; baillif.s@chu-nice.fr (S.B.); drlagier@wanadoo.fr (J.L.); alrabiah-87@hotmail.com (A.A.); martel.a@chu-nice.fr (A.M.); 3Departemnt of Medicine, Cote d’Azur University, F-06000 Nice, France; 4Department of Médicine, Bordeaux University, F-33000 Bordeaux, France

**Keywords:** dacryocystitis, dacryocystorhinostomy, dacryocystectomy, dry eye, epiphora

## Abstract

**Background/Objectives**: The aim of the study was to compare dacryocystectomy (DCT) versus dacryocystorhinostomy (DCR) in patients with dacryocystitis in terms of tearing complaints. **Methods**: We conducted a retrospective and comparative study on 19 patients. The main outcome measure was defined as an improvement by 1 point of the Munk score postoperatively. **Results**: A total of 19 patients were included with 10 in the DCR group and 9 in the DCT group. The primary endpoint was reached in 7 (70%) and in 6 (67%) patients in the DCR and DCT groups, respectively (*p* > 0.999). All DCR procedures were performed under general anesthesia (GA), while almost all DCT procedures were performed under local anesthesia (LA) (*p* < 0.001). There was a higher need for hospitalization in the DCR group (*p* < 0.001). **Conclusions**: Our preliminary results indicate that DCR is not always the solution in the case of dacryocystitis. DCT is a viable surgical procedure, especially in elderly patients without any tearing complaint and with underlying dry eye disease.

## 1. Introduction

Dacryocystitis is a common disorder caused by nasolacrimal duct obstruction (NLDO), leading to inflammation and infection of the lacrimal sac. Two forms of dacryocystitis are defined: acute and chronic. Acute dacryocystitis (AD) is a painful and rapidly progressive infection of the lacrimal sac requiring systemic antibiotics and sometimes surgical drainage. In contrast, chronic dacryocystitis (CD) is a milder and chronic form of the disease that develops gradually and presents with persistent tearing, discharge, and swelling in the medial canthal area (dacryocystocele).

At a distance, dacryocystorhinostomy (DCR) is usually recommended to treat epiphora and avoid AD and CD recurrence [1]. DCR is an invasive procedure consisting of anastomosing the lacrimal sac with the nasal mucosa (Figure 1A). This technique presents several limitations, such as the need for general anesthesia (GA) in our center, the withdrawal of antiplatelets and anticoagulation medications [2], the risk of postoperative dry eye syndrome, and a higher risk of surgical failure in elderly (>80 y-o) patients [3] and in patients with inflammatory diseases involving the nose (e.g., Wegener disease) [4]. In patients with preoperative severe dry eye syndrome (e.g., graft versus host disease), DCR has even been shown to be deleterious [5].

Dacryocystectomy (DCT) was the first lacrimal surgery to be described by Woolhouse in 1724 and consisted of removing the entire lacrimal sac without nasal mucosa anastomosis (Figure 1B) [6]. DCT is mainly performed in lacrimal sac malignancies but is rarely performed in the case of NLDO due to the fear of postoperative epiphora resulting from an interruption of the lacrimal drainage outflow.

However, several studies investigated the use of DCT in chronic NLDO and found that in selected cases, approximately 75% of patients did not complain of any postoperative tearing [7,8,9,10]. In addition, DCT is easily performed under local anesthesia (LA), even in patients under antiplatelet medications. To date, no study compared DCT with DCR in AD or CD patients. Finally, the authors believe that DCT is not taught enough in France. Therefore, the aim of this study was to compare DCT versus DCR in patients referred to our center for AD or CD and to investigate the outcomes with a special focus on postoperative tearing complaints and AD recurrence.

## 2. Materials and Methods

### 2.1. Study Design

A retrospective study was performed on all patients referred to our tertiary care center for AD or CD between January 2020 and August 2022. The diagnosis was based on clinical examination. AD was defined as a lacrimal sac abscess (Figure 2A). CD was defined as a non-inflammatory dilatation of the lacrimal sac with purulent discharge at palpation (Figure 2B). In all cases, the rinsing of the nasolacrimal duct was not permeable. A dacryoscanner was prescribed at the surgeon’s discretion. We included all patients who presented AD or CD. Patients with chronic tearing without any AD or CD were not included. Patients with less than 1 month of follow-up were also excluded. 

### 2.2. Data Collected

Data collected were the following: demographic data, past medical history, clinical presentation, preoperative examination (dry eye syndrome defined as keratitis associated with an evaporative and/or a hyposecretion dryness, lacrimal lake asymmetry, tearing complaint, Munk score, eyelid or lacrimal punctum malposition, keratitis), intraoperative findings, surgical technique used, intraoperative complications (hemorrhage, cerebrospinal fluid leak due to penetration of the cribriform plate), postoperative complications (hemorrhage, infection, persistent tearing, early loss of the silicone tube, and need for additional surgery), postoperative examination at 1 and 3 months postoperatively (Munk score, lacrimal lake asymmetry, type of anesthesia, delay of intubation removal, AD recurrence), duration of follow-up and days of hospitalization. 

### 2.3. Surgical Technique

The decision to perform a DCT or a DCR was at the surgeon’s discretion. As shown in Figure 2, the main indications for performing a DCT rather than a DCR were old age (>80-year-old), patients under anticoagulation or antiplatelets medications for whom discontinuation could be harmful, preoperative dry eye disease or systemic disease associated with high failure rate of the DCR (e.g., Wegener Granulomatosis).

DCT: All the DCT surgeries were performed externally under local anesthesia with sedation or exceptionally under GA. Only anticoagulation medications were withdrawn preoperatively. Antiplatelet medications were not discontinued. The skin was infiltrated with lidocaine 2% and epinephrine 1:200,000 (Xylocaine). A DCR incision (inner canthus) using a 15-Bard-Parker blade was performed. The orbicularis muscle was dissected, and the angular vessels coagulated. The periosteum was incised and released by using a Freer elevator. Further dissection was performed to isolate the medial canthal tendon and the anterior lacrimal crest. The anterior portion of the medial canthal tendon was incised and elevated. The lacrimal sac was sometimes filled with viscoelastic if required and dissected circumferentially down to the nasolacrimal duct inferiorly. The lacrimal sac was removed “en bloc” as much as possible and then sent to the pathologist. Haemostasis was obtained with bipolar cautery. Sometimes, a monocanaliculonasal stent was inserted into the nasolacrimal duct and removed one month later. The incision was closed using interrupted 5–0 Vicryl to approximate the internal part of the wound. The skin was closed with interrupted 5–0 Silk sutures. Antibiotic ointment was prescribed for one week. No systemic antibiotics were prescribed. A surgical video of the procedure is available (https://youtu.be/ATjyraKsV0Q, accessed on 25 February 2023). A QR code is also provided in the Appendix A to access the video. 

DCR: all the procedures were performed under GA with the withdrawal of all blood thinners preoperatively. The nasal cavity was packed with xylocaine and naphazoline mixture. The surgery was performed externally, as described above. Then, the lacrymal bone was fractured, and the osteotomy was enlarged by using rongeurs. The nasal mucosa was then opened allowing the creation of an anterior mucosal flap. A Bowman probe was passed into the lacrimal sac to tent the sac medially, and Westcott scissors were used to open the lacrimal sac from the duct to the fundus, with relaxing incisions at both ends to create the other part of the anterior flap. A portion of the lacrimal sac was sent to the pathologist for routine examination. A bicanaliculonasal silicone stent (BiKa, FCI, Paris, France) was inserted and retrieved intranasally. The 2 anterior flaps were sutured together by using interrupted 6/0 Vicryl sutures. No posterior flaps were performed. 

Closure and postoperative prescriptions were similar to DCT. The bicanalicular stent was removed 3 months postoperatively. No systemic antibiotics were prescribed. 

### 2.4. Outcomes

The main outcome measure was defined as an improvement by 1 point of the Munk score. Munk score was defined as follows:Grade 0: No epiphora.Grade 1: Occasional epiphora requiring wiping less than twice daily.Grade 2: Epiphora requiring two to four wiping per day.Grade 3: Epiphora requiring 5–10 wiping per day.Grade 4: Epiphora requiring >10 wiping or continuous tearing.

Secondary outcomes included the difference in the mean pre and postoperative Munk, lacrimal lake asymmetry, and postoperative complications. 

### 2.5. Ethics

Due to its retrospective design, no ethical approval was required. This study was conducted in accordance with the ethical standards of the institutional and national research committee and with the 1964 Helsinki Declaration and its later amendments or comparable ethical standards.

### 2.6. Statistical Analysis

Numeric variables were expressed as mean (±SD) and discrete outcomes as absolute and relative (%) frequencies. We created 2 groups according to the values of surgery performed (DCR or DCT). Group comparability was assessed by comparing baseline demographic data and follow-up duration between groups. Normality and heteroskedasticity of continuous data were assessed with the Shapiro–Wilk and Levene’s tests, respectively. Continuous outcomes were compared with unpaired Student *t*-test, Welch *t*-test, or Mann–Whitney U test according to data distribution. Discrete outcomes were compared with chi-squared or Fisher’s exact test accordingly. The alpha risk was set to 5%, and two-tailed tests were used. Statistical analysis was performed with EasyMedStat (version 3.21.5; www.easymedstat.com, accessed on 25 February 2023).

## 3. Results

Nineteen patients were included over the study period (Table 1). Of these, 10 were included in the DCR group and 9 in the DCT group. Demographic and preoperative data are shown in Table 1. Females, as well as the involvement of the left eye, were more prevalent. Both groups were similar regarding the presentation (AD or CD) and preoperative keratitis and dry eye syndrome. The mean ASA score was statistically higher in the DCT group (2.78) compared to the DCR group (1.8) (*p* = 0.021).

Table 2 shows the surgical outcomes. The primary endpoint, defined as an improvement by 1 point of Munk score, was reached in 7 (70%) patients in the DCR group and 6 (67%) patients in the DCT group. Based on this primary outcome, no statistical difference was found between the two groups (*p* > 0.999).

Concerning the secondary outcomes, the mean postoperative Munk score was significantly lower in the DCR group (0) compared to the DCT group (0.89) (*p* = 0.033). No difference was found between pre and postoperative lacrimal lake asymmetry between the groups (*p* > 0.999 and *p* > 0.141, respectively). One patient (10%) experienced a stricturotomy in the DCR group. There was one case (11%) of AD recurrence in the DCT group, requiring further surgical revision. All DCR procedures were performed under GA, while almost all DCT procedures were performed under LA (*p* < 0.001). There was a higher need for hospitalization in the DCR group than in the DCT group (*p* < 0.001). 

One surgery was performed under antiplatelet medication in the DCT group without bleeding concerns. The mean follow-up was 4.5 (1–18) months and was comparable between the groups (*p* > 0.999).

## 4. Discussion

Our preliminary results indicate that DCT is a simple and viable surgical procedure for treating AD and CD and may represent an alternative to DCR in selected patients. 

As shown in Table 3, four other studies have previously investigated the impact of DCT on tearing [7,8,9,10]. In 2022, Shadid Alam et al. conducted a prospective and non-comparative study on 68 patients to assess the efficacy of DCT in reducing epiphora in cases of primary acquired NLDO [8]. They found that apart from providing relief from ocular discharge, DCT also provides a significant watering improvement, with a preoperative Munk score of 3.18 vs. 0.67 postoperatively. Three other retrospective studies have assessed the indications, results, and complications of DCT [7,9,10]. They found that approximately 75% of patients did not complain of any postoperative tearing when classifying it as just “present” or “absent”. In our study, we did not find any difference between the two techniques regarding the 1-point improvement in the Munk score. However, there was a slight discrepancy in the mean postoperative Munk score. While statistically significant, suggesting a marginal inferiority of the DCT, we do not consider this difference to be clinically relevant given that it falls within the range of 0 to 1 Munk score, and there was consistent improvement from preoperative to postoperative scores in all cases. None of the patients in the present study complained of significant postoperative tearing requiring surgical revision. Nevertheless, this aspect should be carefully considered when recommending surgery: if tearing is a preoperative concern for the patient, opting for DCR over DCT may be more appropriate to address this issue alongside treating AD or CD.

One of the main fears of DCT is the risk of irreversible postoperative epiphora resulting from an interruption of the lacrimal drainage outflow. However, this “lacrimal plug” effect is particularly suitable in patients with severe dry eye syndrome (e.g., Graft versus Host disease, Gougerot Sjogren syndrome, Lyell syndrome, and post-radiotherapy), increasing the lacrimal meniscus and improving keratoconjunctivitis sicca. In 1996, Boynton and Anawis published three cases of patients with failed DCR, which were then treated successfully with DCT [11]. The authors concluded that DCR, although technically well performed, was associated with a functional failure because these patients had insufficient lacrimal production to maintain the lacrimal flow. In these cases, DCT may become the most appropriate surgical procedure.

However, it can be paradoxical to understand why DCT would be more appropriate for patients with dry eye while it may also improve epiphora in others. Careful preoperative ocular surface dryness is paramount. In patients with dry eye disease, the main concern may not be related to tearing before the surgery but rather the infectious complications (AD) and chronic mucus discharge. By removing the lacrimal sac in these cases, both the dacryocystitis and mucus discharge can be addressed without exacerbating eye dryness by creating a “lacrimal plug” effect. On the other hand, in patients without an underlying dry eye, complaints typically include acute or chronic dacryocystitis, discharge, and sometimes true epiphora. We postulate that the additional epiphora experienced in these cases is due to mucus accumulation in the conjunctival cul de sac, leading to local irritation and reflex tearing. Performing a DCT in these patients resolves the dacryocystitis, mucus discharge, and local irritation, ultimately improving patient comfort in the postoperative period. Another interesting explanation is that the removal of the lacrimal sac may be associated with a tear production decrease through a permanent crosstalk between the lacrimal sac and the lacrimal gland. Although the exact mechanism remains unclear, recent research by Singh et al. [12] suggests a significant communication between the tear drainage and production systems. Their study found that in the case of unilateral lacrimal outflow obstruction, tear flow from the affected lacrimal gland was significantly reduced compared to the unaffected side. This finding implies that the lacrimal drainage system may play a role in regulating tear production, possibly through a feedback mechanism that links the state of tear outflow with tear secretion rates. 

It is also important to discuss the role of lacrimal intubation in DCT, particularly given the absence of the lacrimal apparatus. While this approach may seem counterintuitive, other studies [13] have shown that lacrimal intubation in conjunction with DCT can lead to a significant reduction in postoperative epiphora. In addition, performing lacrimal intubation postoperatively in case of residual tearing was associated with symptom improvements. The authors reported that this procedure was particularly beneficial in well-selected cases, such as elderly patients with compromised lacrimal sacs or those suffering from recurrent infections. Despite these promising findings, the rationale for improved outcomes with lacrimal stenting remains unclear, and the procedure’s overall effectiveness is still a matter of debate. We initially planned to perform a subgroup analysis to assess the impact of lacrimal stenting on the success rates of DCT in our study. Unfortunately, our sample size was insufficient to draw meaningful conclusions. Therefore, further studies with larger patient populations are necessary to fully understand the potential benefits and limitations of this approach in DCT.

In our study, we observed a recurrence of AD in one patient in the DCT group, which required a surgical revision with favorable outcomes. This finding is intriguing, as DCT should theoretically prevent recurrence by removing the entire lacrimal sac. We suspect that this recurrence was due to incomplete excision of the lacrimal sac, with a small remnant persisting and subsequently harboring infection. This underscores the importance of ensuring thorough removal of the sac during DCT to prevent such complications. Recent studies demonstrated that injecting fibrin glue combined with tryptan blue into the lacrimal sac allowed for an easier and safer lacrimal sac excision [14]. This approach distends the sac, facilitating its separation from the surrounding periosteal and fascial attachments. The technique improves the definition of the mucosal lining and aids in achieving an “en bloc” excision of the lacrimal sac without disrupting the fascial plane, ultimately enhancing the precision of the surgery. While it is difficult to predict whether this complication would become more common with a larger cohort, it highlights the need for further study. 

DCR is frequently performed in France [15,16] and is considered the treatment of choice for all patients presenting an AD or CD, regardless of the preoperative examination, including the ocular surface and tearing complaint. However, DCR is an invasive procedure and can be associated with severe complications. In 2017, Tooley and al. investigated the surgical outcomes and complication rates of DCR in an elderly population over 80 years old. They found that elderly patients experienced less symptom resolution compared to younger patients (64% vs. 86%, *p* = 0.02). In addition, the risk of serious complications (stroke, myocardial infarction, hospitalization within 1 month, bleeding requiring intervention, and death) after DCR was higher in the elderly group (15% vs. 2%; *p* = 0.01) [3]. The increased morbidity associated with general anesthesia in this patient population is a key factor to consider when planning surgery. Consequently, in elderly or medically fragile patients, local anesthesia may be preferable to mitigate these risks. While external DCR can be performed under local anesthesia, it is rarely performed for patient comfort. Endonasal DCR, on the other hand, is always performed under general anesthesia. In contrast, DCT is easily managed with local anesthesia. In our study, only two patients in the DCT group underwent general anesthesia; one of these patients was a young individual with Wegener’s granulomatosis.

DCT is rarely mentioned in the literature and is mainly indicated for the resection of lacrimal sac tumors. However, DCT has several advantages over DCR, which are summarized below: DCT can be easily performed under LA with sedation, as outlined in Table 2 and Table 3.DCT can be performed under antiplatelet medications, as stressed by our article. The authors believe that anticoagulants should be withdrawn for bleeding concerns. Additional studies are needed to confirm this assumption.DCT is the treatment of choice in case of severe nasal mucosa involvement, as encountered in Wegener’s granulomatosis [6].DCT is quicker and technically easier compared to DCR because no intranasal manipulation is required.DCT should be considered as a viable surgical alternative to DCR in patients with preoperative dry eye syndrome.As DCT is a quicker procedure, requires a reduced hospitalization stay, and does not necessarily require lacrimal intubation, it could have a lower impact on healthcare costs compared to DCR.

Based on our findings, we established a therapeutic decisional tree for patients with AD or CD based on careful preoperative examination (Figure 3).

Our study is the first to compare DCT versus DCR in AD and CD patients. However, its retrospective design and the small number of patients included require additional studies to confirm our preliminary results. Another limitation of our study is the relatively short mean follow-up period of 4.5 months, which constrains the conclusions that can be drawn regarding long-term outcomes.

## 5. Conclusions

In conclusion, DCR is not always the solution in the case of AD or CD. A careful preoperative examination is mandatory to deliver a personalized surgical treatment. Elderly patients (> 80-year-old) and/or preoperative dry eye syndrome and/or systemic disease involving the nasal mucosa could be an indication of DCT rather than DCR. 

## Figures and Tables

**Figure 1 jcm-13-05129-f001:**
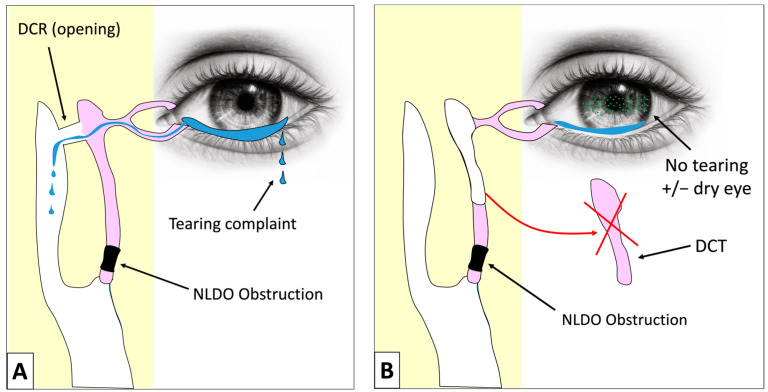
Schematic drawing demonstrating the surgical principle of the DCR (**A**) and the DCT (**B**).

**Figure 2 jcm-13-05129-f002:**
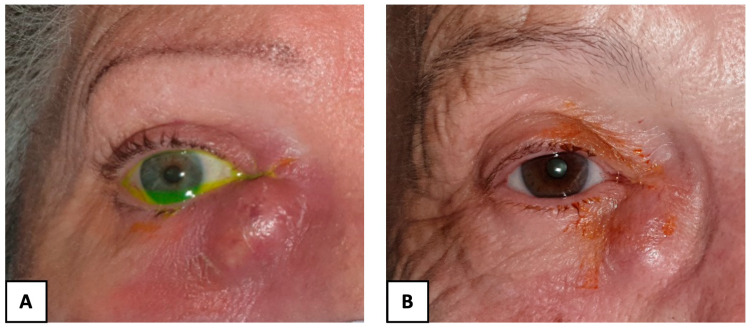
Clinical presentation of acute dacryocycystitis (**A**) and chronic dacryocystitis (**B**).

**Figure 3 jcm-13-05129-f003:**
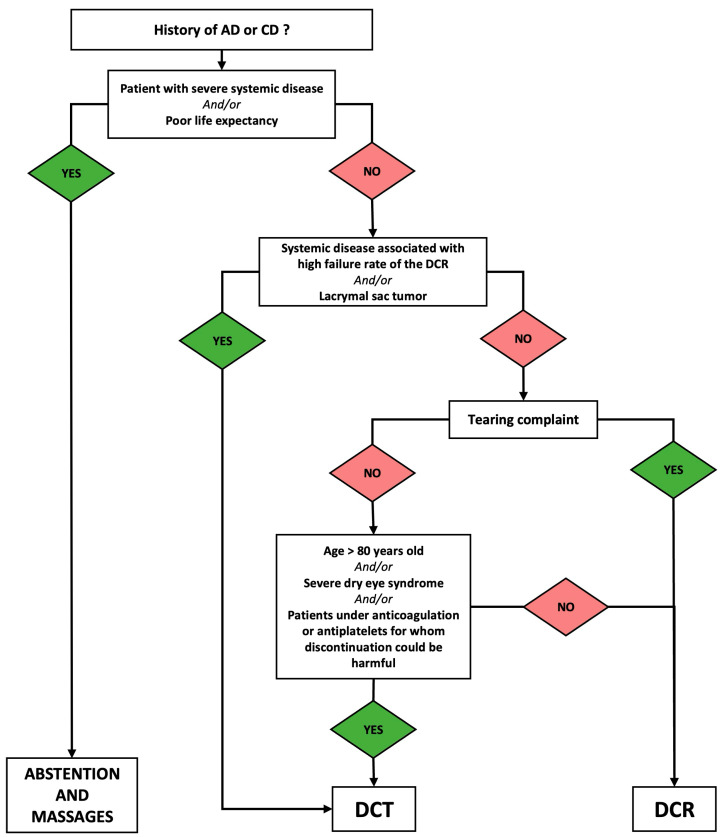
Decision tree of AD or CD treatment based on the preoperative examination.

**Table 1 jcm-13-05129-t001:** Patient demographics.

Characteristics	DCR	DCT	*p*-Value
**Patients, number (%)**	10 (100%)	9 (100%)	>0.999
**Age, mean (range)**	64 (52–78)	74 (44–88)	0.094
**Gender**			
Female, number (%)	9 (90%)	7 (78%)	0.582
Male, number (%)	1 (10%)	2 (22%)	
**Side**			
Left, number (%)	6 (60%)	5 (56%)	>0.999
Right, number (%)	4 (40%)	4 (44%)	
**Clinical presentation**			
Acute dacryocystitis, number (%)	7 (70%)	8 (89%)	>0.999
Number of acute dacryocystitis, mean (range)	1.2 (0–3)	2 (0–4)	0.835
Chronic dacryocystocele, number (%)	3 (30%)	3 (33%)	>0.999
Keratitis, number (%)	1 (10%)	2 (22%)	0.721
Dry eye syndrome, number (%)	1 (10%)	2 (22%)	0.582
**Imaging**			
Dacryoscanner performed, number (%)	7 (70%)	4 (44%)	0.37
NLDO confirmed by scanner, number (%)	7 (70%)	4 (44%)	0.999
Other abnormality, number (%)	1 (10%)	1 (11%)	0.999
**Medical history**			
Arterial hypertension, number (%)	4 (40%)	5 (56%)	0.656
Anticoagulant medication, number (%)	0 (0%)	2 (22%)	0.211
Antiplatelet medication, number (%)	2 (20%)	1 (11%)	>0.999
**ASA score, mean (range)**	1.8 (1–3)	2.78 (2–3)	**0.021**

ASA Score: American Society of Anesthesiologists Score. DCT: Dacryocystectomy. DCR: Dacryocystorhinostomy. NLDO: Naso Lacrymal Duct Obstruction.

**Table 2 jcm-13-05129-t002:** Outcomes.

Outcomes	DCR N = 10	DCT N = 9	*p*-Value
**Grades of Munk scale**			
Pre op, mean (range)	1.8 (0–3)	2 (0–3)	>0.999
Post op, mean (range)	0	0.89 (0–3)	**0.033**
Improvement by 1 point of Munk score, number (%)	7 (70%)	6 (67%)	>0.999
**Lacrimal lake asymmetry**			
Pre op, number (%)	5 (50%)	4 (44%)	>0.999
Post op, number (%)	1 (10%)	4 (44%)	0.141
**Lacrimal intubation**			
Intubation used, number (%)	10 (100%)	5 (56%)	**0.033**
Delay of removal in month, mean (range)	2 (1–3)	1.6 (1–3)	0.404
**Postoperative complications**			
Stricturotomy, number (%)	1 (10%)	0 (0%)	>0.999
Acute or chronic dacryocystitis recurrence, number (%)	0 (0%)	1 (11%)	0.474
**Keratitis**			
Pre op, number (%)	1 (10%)	2 (22%)	0.721
Post op, number (%)	1 (10%)	3 (33%)	0.303
**Types of anesthesia**			
General anesthesia, number (%)	10 (100%)	2 (22%)	**<0.001**
Local anesthesia, number (%)	0 (0%)	7 (78%)	
Type of patient care			
Hospitalization, number (%)	10 (100%)	4 (44%)	**0.011**
Ambulatory care, number (%)	0 (0%)	5 (56%)	
**Surgery performed under blood thinner medications**			
Anticoagulant, number (%)	0 (0%)	0 (0%)	>0.999
Antiplatelet, number (%)	0 (0%)	1 (11%)	0.474
**Follow-up in months, mean (range)**	4.5 (1–12)	4.4 (1–18)	>0.999

DCT: Dacryocystectomy. DCR: Dacryocystorhinostomy.

**Table 3 jcm-13-05129-t003:** Comparison of dacryocystectomy outcomes in different studies.

Authors	Type of Study	Number of Patients	Age, Mean (Range)	Type of Surgery	Pre Op Dry Eye	Pre Op Tearing	Post Op Tearing	Type of Anesthesia	Follow-Up (Month), Mean (Range)
**Mathieu et al.,** **(2023)**	Retrospective and comparative	19	66 (28–88)	DCT and DCR	3 (16%)	Munk 1.9 (0–3)	Munk 0.42 (0–3)	LA (37%) GA (63%)	4.5 (1–18)
**Shadid et al.,** **(2022)**	Prospective, non-comparative	65	68 (60–85)	DCT	N/A	Munk 3.18 (0–3)	Munk 0.67 (0–3)	N/A	1 (0.5–1.5)
**Galindo-Ferreiro et al.,** **(2018)**	Retrospective, non-comparative	47	58 (7–95)	DCT	11 (23.5%)	22 (*) (42.6%)	4 (*) (8.5%)	LA (100%)	12 (1–66)
**Meireles et al.,** **(2017)**	Retrospective, non-comparative	17	76 (68–85)	DCT	N/A	17 (*) (100%)	4 (*) (23.5%)	LA (100%)	N/A
**Matayoshi et al.,** **(2004)**	Retrospective, non-comparative	11	54 (13–93)	DCT	3 (27%)	9 (*) (81%)	1 (*) (9%)	N/A	37 (12–120)

(*): Epiphora classified as “present” or “absent”. DCT: Dacryocystectomy. DCR: Dacryocystorhinostomy. LA: Local anesthesia/GA: General anesthesia. N/A: not applicable.

## Data Availability

The original contributions presented in the study are included in the article/Appendix A. Further inquiries can be directed to the corresponding author.

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
