# Peer review of "Dacryocystitis: Is Dacryocystorhinostomy Always the Solution?"

_jcm, 2024, doi:10.3390/jcm13175129_

Round 1

Reviewer 1 Report

Comments and Suggestions for Authors

The current manuscript aims to compare dacryocystectomy (DCT) versus dacryocystorhinostomy (DCR) in patients with dacryocystitis in terms of tearing complaint. Although the topic is interesting in its scientific field, there are some issues that require the authors’ attention to improve the quality of this particular manuscript before further consideration for publication in a high-quality journal “JCM”.

Specific comments:

1.         What are the long-term recurrence rates of dacryocystitis in patients treated with dacryocystectomy (DCT) versus dacryocystorhinostomy (DCR)? Please specify.

2.         According to the authors’ statement, the mean postoperative Munk score (0) in the DCR group was significantly lower than that in the DCT group (0.89), p=0.033. However, they did not discuss the clinical significance of this difference in postoperative epiphora control. Please improve. Furthermore, whether it might affect the choice of surgery? Please justify.

3.         Please clarify the role of using a lacrimal stent in the success rates of DCT and DCR. Furthermore, whether it might affect the postoperative epiphora and other symptoms of dacryocystitis recurrence? Please justify.

4.         What are the differences in surgical success rates and complication rates between DCR and DCT in patients of different age groups? Are there certain patient groups (such as the elderly or those with dry eye syndrome) who are at lower risk of complications with DCT? Please clarify.

5.         During DCR and DCT surgery, how the anesthetic manner (general anesthesia versus local anesthesia) may affect surgical outcomes and patient recovery? Please justify.

Author Response

ANSWER TO REVIEWER 1

Dear reviewer,

Thank you sincerely for your valuable and constructive feedback. We have addressed all the points raised in your correspondence. Should you have any further suggestions for improvement, please do not hesitate to let us know.

Best regards,
The corresponding author

1st Remark:

What are the long-term recurrence rates of dacryocystitis in patients treated with dacryocystectomy (DCT) versus dacryocystorhinostomy (DCR)? Please specify.

Response:

You are correct to ask about the long-term recurrence rates, as this is a crucial consideration. In our study, we observed one case of AD recurrence after a DCT. This recurrence occurred 6 months after the surgery and required another DCT with favorable outcomes. This finding is intriguing, as DCT should theoretically prevent recurrence by completely removing the lacrimal sac. We hypothesize that this recurrence was due to incomplete excision of the lacrimal sac, with a small remnant persisting and subsequently harboring infection. This underscores the importance of ensuring thorough removal of the sac during DCT to prevent such complications.

Since writing this article, we now inject fibrin glue into the lacrimal sac to ensure total excision of the lacrimal sac. This technique has been described in detail by Jessica Y. Tong and Dinesh Selva in their 2023 study "Dacryocystectomy: A Fibrin Glue-Assisted Subfascial Excision" (Indian J Ophthalmol, 71(5):2260-2262). It involves the injection of Tisseel fibrin glue mixed with trypan blue into the lacrimal sac. This approach distends the sac, facilitating its separation from the surrounding periosteal and fascial attachments. The technique improves the definition of the mucosal lining and aids in achieving an “en bloc” excision of the lacrimal sac without disrupting the fascial plane, ultimately enhancing the precision of the surgery.

We added a paragraph in the discussion from line 237 to line 252 to address this point.

We have detailed this recurrence rate in the results section of our manuscript (line 162). Unfortunately, another limitation of our study is the relatively short mean follow-up period of 4.5 months, which constrains the conclusions that can be drawn regarding long-term outcomes. This has been added in the discussion of the revised manuscript line 292-294. In our experience until now, none of the patients who have undergone DCT or DCR surgery in this study complained of significant tearing requiring surgical revision.

2nd Remark:

According to the authors’ statement, the mean postoperative Munk score (0) in the DCR group was significantly lower than that in the DCT group (0.89), p=0.033. However, they did not discuss the clinical significance of this difference in postoperative epiphora control. Please improve. Furthermore, whether it might affect the choice of surgery? Please justify.

Response:

Thank you for pointing out this important consideration. While statistically significant, suggesting a marginal inferiority of the DCT, we do not consider this difference to be clinically relevant given that it falls within the range of 0 to 1 Munk score, and there was consistent improvement from pre-operative to post-operative scores in all cases. Until now, none of the patients undergoing DCT complained of significant tearing requiring surgical revision.

Nevertheless, this aspect should be carefully considered when recommending surgery: if tearing is a preoperative concern for the patient, a DCR surgery appears more appropriate to address this issue alongside treating AD or CD. We have developed this point in lines 183 to 192 of our revised manuscript.

3rd Remark:

Please clarify the role of using a lacrimal stent in the success rates of DCT and DCR. Furthermore, whether it might affect the postoperative epiphora and other symptoms of dacryocystitis recurrence? Please justify.

Response:

You raise an interesting point regarding the role of lacrimal stenting. We acknowledge that intubating a missing lacrimal apparatus appears counterintuitive. However, other studies such as the one of Nuzzi and al. (Dacryocystectomy with Lacrimal Silicone Intubation in Challenging Patients Affected by Recurrent Dacryocystitis and Epiphora: Expanding Minimally Invasive Approach Indications) have shown that lacrimal intubation in conjunction with DCT can lead to a significant reduction in postoperative epiphora. The authors reported that this procedure was particularly beneficial in well-selected cases, such as elderly patients with compromised lacrimal sacs or those suffering from recurrent infections. In addition, previous studies found that intubation the lacrimal apparatus following a DCT could lead to an epiphora decrease in several patients. Clearly, the rationale for improved outcomes with lacrimal stenting remains unclear, and the procedure’s overall effectiveness is still a matter of debate. One may hypothesize that an orbital drainage might be favored postoperatively. A lack of statistical power in the previous studies cannot be excluded. For example, our sample size was not enough to conduct a subgroup analysis regarding lacrimal intubation or not in the DCT group. Therefore, further studies with larger patient populations are necessary to fully understand the potential benefits and limitations of this approach in DCT.

We have also added a section in the Discussion from line 224 to 236 to address the debated of this intubation approach.

4th Remark:

What are the differences in surgical success rates and complication rates between DCR and DCT in patients of different age groups? Are there certain patient groups (such as the elderly or those with dry eye syndrome) who are at lower risk of complications with DCT? Please clarify.

Response:

Thank you for highlighting this issue. While we did not find significant differences in surgical success or complication rates between DCR and DCT across different age groups in our study, the sample size was too limited to perform a robust subgroup analysis. Although not significant, patients in the DCT group tended to be older compared to the patients included in the DCT group (see Table 1). Based on existing literature and our observations, patients with underlying dry eye syndrome may experience exacerbation of symptoms following DCR, as the procedure can increase lacrimal outflow, potentially aggravating ocular surface dryness. This is an area we hope to investigate further in future studies with a larger patient population. Currently, our data do not allow us to draw definitive conclusions due to the limited number of cases.

5th Remark:

During DCR and DCT surgery, how the anesthetic manner (general anesthesia versus local anesthesia) may affect surgical outcomes and patient recovery? Please justify.

Response:

You are right to inquire about the impact of anesthesia on surgical outcomes and recovery. While we do not believe that the choice between general and local anesthesia significantly affects the surgical outcome, it can have a notable impact on patient recovery. As evidenced by the study conducted by Tooley et al., elderly patients undergoing DCR are at a higher risk of serious complications, such as stroke, myocardial infarction, and even death, within the first postoperative month. The increased morbidity associated with general anesthesia in this patient population is a key factor to consider when planning surgery. Consequently, in elderly or medically fragile patients, local anesthesia may be preferable to mitigate these risks. While external DCR can be performed under local anesthesia, it is rarely done so for patient comfort. Endonasal DCR, on the other hand, is always performed under general anesthesia. In contrast, DCT is easily managed with local anesthesia. In our study, only two patients in the DCT group underwent general anesthesia; one of these patients was a young individual with Wegener's granulomatosis.

We added a paragraph from line 262 to line 269 to develop this matter.

Reviewer 2 Report

Comments and Suggestions for Authors

It would seem that the main effect of a DCT procedure would be to prevent further AD or CD. In this respect, the fact that there was one case of AD recurrence (11%) in the DCT group may become significant should the study group become larger and the percentage of postop complications remain the same.

However, the primary endpoint is to improve the Munk score. I would like the authors to discuss what would be the mechanism for this improvement. It is unclear why DCT would be more appropriate for patients with dry eye while it may improve the epiphora in others.

56% of DCT patients received an intubation. What were the criteria for using it? Could the use be related to the improvement of the Munk score?

Also, the percentage of patients that developed a keratitis was quite high in the DCT group. Was it infectious keratitis? 

Author Response

ANSWER TO REVIEWER 2

Dear reviewer,

Thank you sincerely for your valuable and constructive feedback. We have addressed all the points raised in your correspondence. Should you have any further suggestions for improvement, please do not hesitate to let us know.

Best regards,
The corresponding author

1st Remark:

It would seem that the main effect of a DCT procedure would be to prevent further AD or CD. In this respect, the fact that there was one case of AD recurrence (11%) in the DCT group may become significant should the study group become larger, and the percentage of postop complications remain the same.

Response:

You are correct to ask about the long-term recurrence rates, particularly if the study cohort expands. In our study, we observed a recurrence of acute dacryocystitis in one patient within the DCT group, which required further surgical intervention. This finding is intriguing, as DCT should theoretically prevent recurrence by completely removing the lacrimal sac. We hypothesize that this recurrence was due to incomplete excision of the lacrimal sac, with a small remnant persisting and subsequently harboring infection. This underscores the importance of ensuring thorough removal of the sac during DCT to prevent such complications.

Since writing this article, we have adopted a technique where fibrin glue is injected into the lacrimal sac to ensure a more precise excision. This method, as described by Jessica Y. Tong and Dinesh Selva in their 2023 study "Dacryocystectomy: A Fibrin Glue-Assisted Subfascial Excision" (Indian J Ophthalmol, 71(5):2260-2262). It involves the injection of Tisseel fibrin glue mixed with trypan blue into the lacrimal sac. This approach distends the sac, facilitating its separation from the surrounding periosteal and fascial attachments. The technique improves the definition of the mucosal lining and aids in achieving an “en bloc” excision of the lacrimal sac without disrupting the fascial plane, ultimately enhancing the precision of the surgery.

While it’s difficult to predict whether this complication would become more common with a larger cohort, it highlights the need for further study. We aim to address this question more thoroughly as our patient cohort expands.

We added a paragraph in the discussion from line 237 to line 252 to address this point.

2nd Remark:

However, the primary endpoint is to improve the Munk score. I would like the authors to discuss what would be the mechanism for this improvement. It is unclear why DCT would be more appropriate for patients with dry eye while it may improve the epiphora in others.

Response:

Thank you for bringing up this important point. As you mentioned, it is crucial to distinguish between patients with dry eye and those without. In patients with dry eye, their main concern may not be related to tearing before the surgery, but rather to issues such as dacryocystitis or mucus discharge instead of true epiphora. By removing the lacrimal sac in these cases, both the dacryocystitis and mucus discharge can be addressed without exacerbating eye dryness, essentially creating a "lacrimal plug" effect.

On the other hand, in patients without underlying dry eye, complaints typically include dacryocystitis, discharge, and sometimes true epiphora. We postulate that the additional epiphora experienced in these cases is due to mucus accumulation in the conjunctival cul de sac, leading to local irritation and reflex tearing. Conducting a DCT in these patients resolves the dacryocystitis, mucus discharge, and local irritation, ultimately improving patient comfort in the post-operative period.

Another interesting explanation is that the removal of the lacrimal sac may be associated with a decrease of tears production through a crosstalk between the lacrimal sac and the lacrimal gland. Although the exact mechanism remains unclear, recent research by Singh et al. ("Lacrimal gland activity in lacrimal drainage obstruction: exploring the potential cross-talk between the tear secretion and outflow," Br J Ophthalmol, 2024) suggests a significant communication between the tear drainage and production systems. Their study found that in cases of unilateral lacrimal outflow obstruction, tear flow from the affected lacrimal gland was significantly reduced compared to the unaffected side. This finding implies that the lacrimal drainage system may play a role in regulating tear production, possibly through a feedback mechanism that links the state of tear outflow with tear secretion rates. In daily clinical practice, we have all been faced to patients with removal of lacrimal apparatus for eyelid cancers and even lacrimal sac cancers who did not complain of any epiphora postoperatively. Further studies are warranted to explore this suspected crosstalk.

We agree with you that it is crucial to clarify this point in the manuscript. Therefore, we have included a section from line 203 to line 223 for further explanations.

3rd Remark:

56% of DCT patients received an intubation. What were the criteria for using it? Could the use be related to the improvement of the Munk score?

Response:

You raise an interesting point regarding the role of lacrimal stenting. We acknowledge that intubating a missing lacrimal apparatus appears counterintuitive. However, other studies such as the one of Nuzzi and al. (Dacryocystectomy with Lacrimal Silicone Intubation in Challenging Patients Affected by Recurrent Dacryocystitis and Epiphora: Expanding Minimally Invasive Approach Indications) have shown that lacrimal intubation in conjunction with DCT can lead to a significant reduction in postoperative epiphora. The authors reported that this procedure was particularly beneficial in well-selected cases, such as elderly patients with compromised lacrimal sacs or those suffering from recurrent infections. In addition, previous studies found that intubation the lacrimal apparatus following a DCT could lead to an epiphora decrease in several patients. Clearly, the rationale for improved outcomes with lacrimal stenting remains unclear, and the procedure’s overall effectiveness is still a matter of debate. One may hypothesize that an orbital drainage might be favored postoperatively. A lack of statistical power in the previous studies cannot be excluded. For example, our sample size was not enough to conduct a subgroup analysis regarding lacrimal intubation or not in the DCT group. Therefore, further studies with larger patient populations are necessary to fully understand the potential benefits and limitations of this approach in DCT.

We have also added a section in the Discussion from line 224 to 236 to address the debated of this intubation approach.

4th Remark:

Also, the percentage of patients that developed a keratitis was quite high in the DCT group. Was it infectious keratitis? 

Response:

Thank you for bringing attention to this matter. The disparity in keratitis cases between the DCT and DCR groups is primarily influenced by our selection criteria for recommending either procedure to a patient. Specifically, when a patient presents with underlying dry eye disease, we tend to lean towards DCT for the reasons previously mentioned. Consequently, since there were more patients with underlying dry eye disease in the DCT group, there was also a higher incidence of keratitis both pre- and post-operatively within this group. It's worth noting that the keratitis cases observed were non-infectious here.

Round 2

Reviewer 1 Report

Comments and Suggestions for Authors

The revised version has adequately addressed most of the critiques raised by this reviewer and is now suitable for publication in "JCM".